# Unsupervised Scalable Representation Learning for Multivariate Time Series

**Jean-Yves Franceschi**[*]
Sorbonne Université, CNRS, LIP6, F-75005 Paris, France
`jean-yves.franceschi@lip6.fr`

**Aymeric Dieuleveut**
MLO, EPFL, Lausanne CH-1015, Switzerland
CMAP, Ecole Polytechnique, Palaiseau, France
`aymeric.dieuleveut@polytechnique.edu`

**Martin Jaggi**
MLO, EPFL, Lausanne CH-1015, Switzerland
`martin.jaggi@epfl.ch`

## Abstract

Time series constitute a challenging data type for machine learning algorithms, due to their highly variable lengths and sparse labeling in practice. In this paper, we tackle this challenge by proposing an unsupervised method to learn universal embeddings of time series. Unlike previous works, it is scalable with respect to their length and we demonstrate the quality, transferability and practicability of the learned representations with thorough experiments and comparisons. To this end, we combine an encoder based on causal dilated convolutions with a novel triplet loss employing time-based negative sampling, obtaining general-purpose representations for variable length and multivariate time series.

## 1 Introduction

We investigate in this work the topic of unsupervised general-purpose representation learning for time series. In spite of the increasing amount of work about representation learning in fields like natural language processing (Young et al., 2018) or videos (Denton & Birodkar, 2017), few articles explicitly deal with general-purpose representation learning for time series without structural assumption on non-temporal data.

This problem is indeed challenging for various reasons. First, real-life time series are rarely or sparsely labeled. Therefore, *unsupervised* representation learning would be strongly preferred. Second, methods need to deliver compatible representations while allowing the input time series to have unequal lengths. Third, scalability and efficiency both at training and inference time is crucial, in the sense that the techniques must work for both short and long time series encountered in practice.

Hence, we propose in the following an *unsupervised* method to learn *general-purpose representations* for *multivariate* time series that comply with the issues of *varying and potentially high lengths* of the studied time series. To this end, we introduce a novel unsupervised loss training a scalable encoder, shaped as a deep convolutional neural network with dilated convolutions (Oord et al., 2016) and outputting fixed-length vector representations regardless of the length of its output. This loss is built

---

[*]Work partially done while studying at ENS de Lyon and MLO, EPFL.

as a triplet loss employing time-based negative sampling, taking advantage of the encoder resilience to time series of unequal lengths. To our knowledge, it is the first fully unsupervised triplet loss in the literature of time series.

We assess the quality of the learned representations on various datasets to ensure their universality. In particular, we test how our representations can be used for classification tasks on the standard datasets in the time series literature, compiled in the UCR repository (Dau et al., 2018). We show that our representations are *general* and *transferable*, and that our method *outperforms concurrent unsupervised methods* and even *matches the state of the art* of non-ensemble supervised classification techniques. Moreover, since UCR time series are exclusively univariate and mostly short, we also evaluate our representations on the recent UEA multivariate time series repository (Bagnall et al., 2018), as well as on a real-life dataset including very long time series, on which we demonstrate *scalability*, *performance* and generalization ability *across different tasks* beyond classification.

This paper is organized as follows. Section 2 outlines previous works on unsupervised representation learning, triplet losses and deep architectures for time series in the literature. Section 3 describes the unsupervised training of the encoder, while Section 4 details the architecture of the latter. Finally, Section 5 provides results of the experiments that we conducted to evaluate our method.

## 2  Related Work

**Unsupervised learning for time series.**   To our knowledge, apart from those dealing with videos or high-dimensional data (Srivastava et al., 2015; Denton & Birodkar, 2017; Villegas et al., 2017; Oord et al., 2018), few recent works tackle unsupervised representation learning for time series. Fortuin et al. (2019) deal with a related but different problem to this work, by learning temporal representations of time series that represent well their evolution. Hyvarinen & Morioka (2016) learn representations on evenly sized subdivisions of time series by learning to discriminate between those subdivisions from these representations. Lei et al. (2017) expose an unsupervised method designed so that the distances between learned representations mimic a standard distance (Dynamic Time Warping, DTW) between time series. Malhotra et al. (2017) design an encoder as a recurrent neural network, jointly trained with a decoder as a sequence-to-sequence model to reconstruct the input time series from its learned representation. Finally, Wu et al. (2018a) compute feature embeddings generated in the approximation of a carefully designed and efficient kernel.

However, these methods either are not scalable nor suited to long time series (due to the sequential nature of a recurrent network, or to the use of DTW with a quadratic complexity with respect to the input length), are tested on no or very few standard datasets and with no publicly available code, or do not provide sufficient comparison to assess the quality of the learned representations. Our scalable model and extensive analysis aim at overcoming these issues, besides outperforming these methods.

**Triplet losses.**   Triplet losses have recently been widely used in various forms for representation learning in different domains (Mikolov et al., 2013; Schroff et al., 2015; Wu et al., 2018b) and have also been theoretically studied (Arora et al., 2019), but have not found much use for time series apart from audio (Bredin, 2017; Lu et al., 2017; Jansen et al., 2018), and never, to our knowledge, in a fully unsupervised setting, as existing works assume the existence of class labels or annotations in the training data. Closer to our work even though focusing on a different, more specific task, Turpault et al. (2019) learn audio embeddings in a semi-supervised setting, while partially relying on specific transformations of the training data to sample positive samples in the triplet loss; Logeswaran & Lee (2018) train a sentence encoder to recognize, among randomly chosen sentences, the true context of another sentence, which is a difficult method to adapt to time series. Our method instead relies on a more natural choice of positive samples, learning similarities using subsampling.

**Convolutional networks for time series.**   Deep convolutional neural networks have recently been successfully applied to time series classification tasks (Cui et al., 2016; Wang et al., 2017), showing competitive performance. Dilated convolutions, popularized by WaveNet (Oord et al., 2016) for audio generation, have been used to improve their performance and were shown to perform well as sequence-to-sequence models for time series forecasting (Bai et al., 2018) using an architecture that inspired ours. These works particularly show that dilated convolutions help to build networks for sequential tasks outperforming recurrent neural networks in terms of both efficiency and prediction performance.

# 3 Unsupervised Training

We seek to train an encoder-only architecture, avoiding the need to jointly train with a decoder as in autoencoder-based standard representation learning methods as done by Malhotra et al. (2017), since those would induce a larger computational cost. To this end, we introduce a novel triplet loss for time series, inspired by the successful and by now classic word representation learning method known as word2vec (Mikolov et al., 2013). The proposed triplet loss uses original time-based sampling strategies to overcome the challenge of learning on unlabeled data. As far as we know, this work is the first in the time series literature to rely on a triplet loss in a fully unsupervised setting.

The objective is to ensure that similar time series obtain similar representations, with no supervision to learn such similarity. Triplet losses help to achieve the former (Schroff et al., 2015), but require to provide pairs of similar inputs, thus challenging the latter. While previous supervised works for time series using triplet losses assume that data is annotated, we introduce an unsupervised time-based criterion to select pairs of similar time series and taking into account

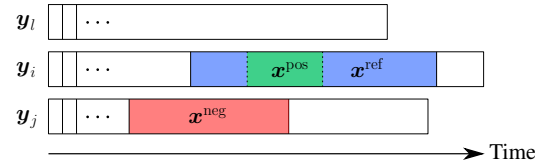

Figure 1: Choices of $\boldsymbol{x}^{\mathrm{ref}}$, $\boldsymbol{x}^{\mathrm{pos}}$ and $\boldsymbol{x}^{\mathrm{neg}}$.

time series of varying lengths, by following word2vec's intuition. The assumption made in the CBOW model of word2vec is twofold. The representation of the *context* of a word should probably be, on one hand, close to the one of this word (Goldberg & Levy, 2014), and, on the other hand, distant from the one of randomly chosen words, since they are probably unrelated to the original word's context. The corresponding loss then pushes pairs of (context, word) and (context, random word) to be linearly separable. This is called *negative sampling*.

To adapt this principle to time series, we consider (see Figure 1 for an illustration) a random subseries[2] $\boldsymbol{x}^{\mathrm{ref}}$ of a given time series $\boldsymbol{y}_i$. Then, on one hand, the representation of $\boldsymbol{x}^{\mathrm{ref}}$ should be close to the one of any of its subseries $\boldsymbol{x}^{\mathrm{pos}}$ (a *positive* example). On the other hand, if we consider another subseries $\boldsymbol{x}^{\mathrm{neg}}$ (a *negative* example) chosen at random (in a different random time series $\boldsymbol{y}_j$ if several series are available, or in the same time series if it is long enough and not stationary), then its representation should be distant from the one of $\boldsymbol{x}^{\mathrm{ref}}$. Following the analogy with word2vec, $\boldsymbol{x}^{\mathrm{pos}}$ corresponds to a word, $\boldsymbol{x}^{\mathrm{ref}}$ to its context, and $\boldsymbol{x}^{\mathrm{neg}}$ to a random word. To improve the stability and convergence of the training procedure as well as the experimental results of our learned representations, we introduce, as in word2vec, several negative samples $(\boldsymbol{x}_k^{\mathrm{neg}})_{k \in [\![1,K]\!]}$, chosen independently at random.

The objective corresponding to these choices to minimize during training can be thought of the one of word2vec with its shallow network replaced by a deep network $\boldsymbol{f}(\cdot, \boldsymbol{\theta})$ with parameters $\boldsymbol{\theta}$, or formally

$$-\log\Big(\sigma\Big(\boldsymbol{f}\big(\boldsymbol{x}^{\mathrm{ref}}, \boldsymbol{\theta}\big)^{\top} \boldsymbol{f}\big(\boldsymbol{x}^{\mathrm{pos}}, \boldsymbol{\theta}\big)\Big)\Big) - \sum_{k=1}^{K} \log\Big(\sigma\Big(-\boldsymbol{f}\big(\boldsymbol{x}^{\mathrm{ref}}, \boldsymbol{\theta}\big)^{\top} \boldsymbol{f}(\boldsymbol{x}_k^{\mathrm{neg}}, \boldsymbol{\theta})\Big)\Big), \qquad (1)$$

where $\sigma$ is the sigmoid function. This loss pushes the computed representations to distinguish between $\boldsymbol{x}^{\mathrm{ref}}$ and $\boldsymbol{x}^{\mathrm{neg}}$, and to assimilate $\boldsymbol{x}^{\mathrm{ref}}$ and $\boldsymbol{x}^{\mathrm{pos}}$. Overall, the training procedure consists in traveling through the training dataset for several epochs (possibly using mini-batches), picking tuples $\big(\boldsymbol{x}^{\mathrm{ref}}, \boldsymbol{x}^{\mathrm{pos}}, (\boldsymbol{x}_k^{\mathrm{neg}})_k\big)$ at random as detailed in Algorithm 1, and performing a minimization step on the corresponding loss for each pair, until training ends. The overall computational and memory cost is $\mathcal{O}(K \cdot c(\boldsymbol{f}))$, where $c(\boldsymbol{f})$ is the cost of evaluating and backpropagating through $\boldsymbol{f}$ on a time series; thus this unsupervised training is scalable as long as the encoder architecture is scalable as well.

The length of the negative examples is chosen at random in Algorithm 1 for the most general case; however, their length can also be the same for all samples and equal to $\mathrm{size}(\boldsymbol{x}^{\mathrm{pos}})$. The latter case is suitable when all time series in the dataset have equal lengths, and speeds up the training procedure thanks to computation factorizations; the former case is only used when time series in the dataset do not have the same lengths, as we experimentally saw no other difference than time efficiency between the two cases. In our experiments, we do not cap the lengths of $\boldsymbol{x}^{\mathrm{ref}}$, $\boldsymbol{x}^{\mathrm{pos}}$ and $\boldsymbol{x}^{\mathrm{neg}}$ since they are already limited by the length of the train time series, which corresponds to scales of lengths on which our representations are tested.

**Algorithm 1:** Choices of $\boldsymbol{x}^{\mathrm{ref}}$, $\boldsymbol{x}^{\mathrm{pos}}$ and $(\boldsymbol{x}_k^{\mathrm{neg}})_{k \in [\![1,K]\!]}$ for an epoch over the set $(\boldsymbol{y}_i)_{i \in [\![1,N]\!]}$.

---

1  **for** $i \in [\![1, N]\!]$ **with** $s_i = \mathrm{size}(\boldsymbol{y}_i)$ **do**
2       pick $s^{\mathrm{pos}} = \mathrm{size}(\boldsymbol{x}^{\mathrm{pos}})$ in $[\![1, s_i]\!]$ and $s^{\mathrm{ref}} = \mathrm{size}(\boldsymbol{x}^{\mathrm{ref}})$ in $[\![s^{\mathrm{pos}}, s_i]\!]$ uniformly at random;
3       pick $\boldsymbol{x}^{\mathrm{ref}}$ uniformly at random among subseries of $\boldsymbol{y}_i$ of length $s^{\mathrm{ref}}$;
4       pick $\boldsymbol{x}^{\mathrm{pos}}$ uniformly at random among subseries of $\boldsymbol{x}^{\mathrm{ref}}$ of length $s^{\mathrm{pos}}$;
5       pick uniformly at random $i_k \in [\![1, N]\!]$, then $s_k^{\mathrm{neg}} = \mathrm{size}(\boldsymbol{x}_k^{\mathrm{neg}})$ in $[\![1, \mathrm{size}(\boldsymbol{y}_k)]\!]$ and finally $\boldsymbol{x}_k^{\mathrm{neg}}$ among subseries of $\boldsymbol{y}_k$ of length $s_k^{\mathrm{neg}}$, for $k \in [\![1, K]\!]$.

---

We highlight that this time-based triplet loss leverages the ability of the chosen encoder to take as input time series of different lengths. By training the encoder on a range of input lengths going from one to the length of the longest time series in the train set, it becomes able to output meaningful and transferable representations regardless of the input length, as shown in Section 5.

This training procedure is interesting in that it is efficient enough to be run over long time series (see Section 5) with a scalable encoder (see Section 4), thanks to its decoder-less design and the separability of the loss, on which a backpropagation per term can be performed to save memory.[3]

## 4  Encoder Architecture

We explain and present in this section our choice of architecture for the encoder, which is motivated by three requirements: it must extract relevant information from time series; it needs to be time- and memory-efficient, both for training and testing; and it has to allow variable-length inputs. We choose to use deep neural networks with *exponentially dilated causal convolutions* to handle time series. While they have been popularized in the context of sequence generation (Oord et al., 2016), they have never been used for unsupervised time series representation learning. They offer several advantages.

Compared to recurrent neural networks, which are inherently designed for sequence-modeling tasks and thus sequential, these networks are scalable as they allow efficient parallelization on modern hardware such as GPUs. Besides this demonstrated efficiency, exponentially dilated convolutions have also been introduced to better capture, compared to full convolutions, long-range dependencies at constant depth by exponentially increasing the receptive field of the network (Oord et al., 2016; Yu & Koltun, 2016; Bai et al., 2018).

Convolutional networks have also been demonstrated to be performant on various aspects for sequential data. For instance, recurrent networks are known to be subject to the issue of exploding and vanishing gradients, due to their recurrent nature (Goodfellow et al., 2016, Chapter 10.9). While significant work has been done to tackle this issue and improve their ability to capture long-term dependencies, such as the LSTM (Hochreiter & Schmidhuber, 1997), recurrent networks are still outperformed by convolutional networks on this aspect (Bai et al., 2018). On the specific domains of time series classification, which is an essential part of our experimental evaluation, and forecasting, deep neural networks have recently been successfully used (Bai et al., 2018; Ismail Fawaz et al., 2019).

Our model is particularly based on stacks of dilated *causal* convolutions (see Figure 2a), which map a sequence to a sequence of the same length, such that the $i$-th element of the output sequence is computed using only values up until the $i$-th element of the input sequence, for all $i$. It is thus called causal, since the output value corresponding to a given time step is not computed using future input values. Causal convolutions allow alleviating the disadvantage of not using recurrent networks at testing time. Indeed, recurrent networks can be used in an online fashion, thus saving memory and computation time during testing. In our case, causal convolutions organize the computational graph so that, in order to update its output when an element is added at the end of the input time series, one only has to evaluate the highlighted graph shown in Figure 2a rather than the full graph.

Inspired by Bai et al. (2018), we build each layer of our network as a combination of causal convolutions, weight normalizations (Salimans & Kingma, 2016), leaky ReLUs and residual connections (see Figure 2b). Each of these layers is given an exponentially increasing dilation parameter ($2^i$ for the

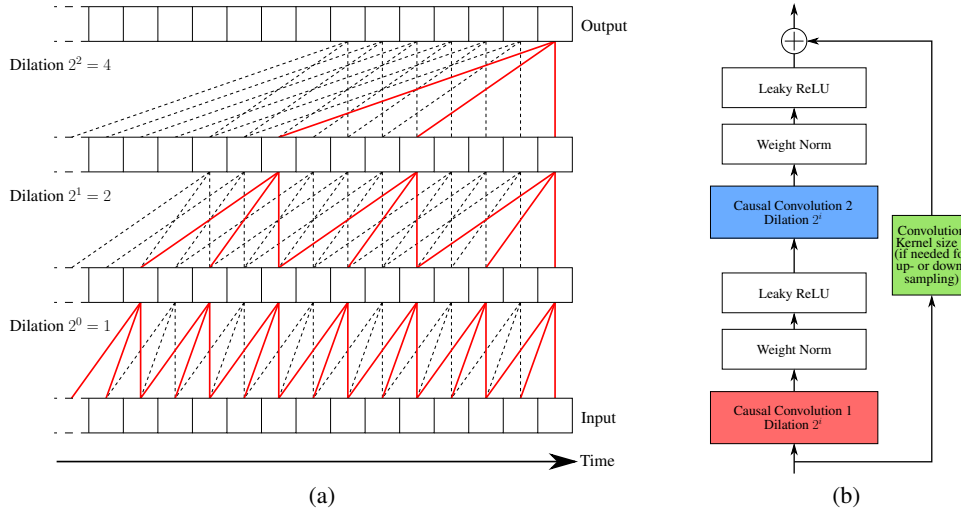

Figure 2: (a) Illustration of three stacked dilated causal convolutions. Lines between each sequence represent their computational graph. Red solid lines highlight the dependency graph for the computation of the last value of the output sequence, showing that no future value of the input time series is used to compute it. (b) Composition of the $i$-th layer of the chosen architecture.

$i$-th layer). The output of this causal network is then given to a global max pooling layer squeezing the temporal dimension and aggregating all temporal information in a fixed-size vector (as proposed by Wang et al. (2017) in a supervised setting with full convolutions). A linear transformation of this vector is then the output of the encoder, with a fixed, independent from the input length, size.

## 5  Experimental Results

We review in this section experiments conducted to investigate the relevance of the learned representations. The code corresponding to these experiments is attached in the supplementary material and is publicly available.[4] The full training process and hyperparameter choices are detailed in the supplementary material, Sections S1 and S2. We used Python 3 for implementation, with PyTorch 0.4.1 (Paszke et al., 2017) for neural networks and scikit-learn (Pedregosa et al., 2011) for SVMs. Each encoder was trained using the Adam optimizer (Kingma & Ba, 2015) on a single Nvidia Titan Xp GPU with CUDA 9.0, unless stated otherwise.

Selecting hyperparameters for an unsupervised method is challenging since the plurality of downstream tasks is usually supervised. Therefore, as Wu et al. (2018a), we choose for each considered dataset archive a single set of hyperparameters regardless of the downstream task. Moreover, we highlight that we perform *no hyperparameter optimization* of the unsupervised encoder architecture and training parameters for any task, unlike other unsupervised works such as TimeNet (Malhotra et al., 2017). Particularly, for classification tasks, *no label* was used during the encoder training.

### 5.1  Classification

We first assess the *quality* of our learned representations on supervised tasks in a standard manner (Xu et al., 2003; Dosovitskiy et al., 2014) by using them for time series classification. In this setting, we show that (1) our method outperforms state-of-the-art unsupervised methods, and notably achieves performance close to the supervised state of the art, (2) strongly outperforms supervised deep learning methods when data is only sparsely labeled, (3) produces transferable representations.

For each considered dataset with a train / test split, we unsupervisedly train an encoder using its train set. We then train an SVM with radial basis function kernel on top of the learned features using the train labels of the dataset, and output the corresponding classification score on the test set. As our

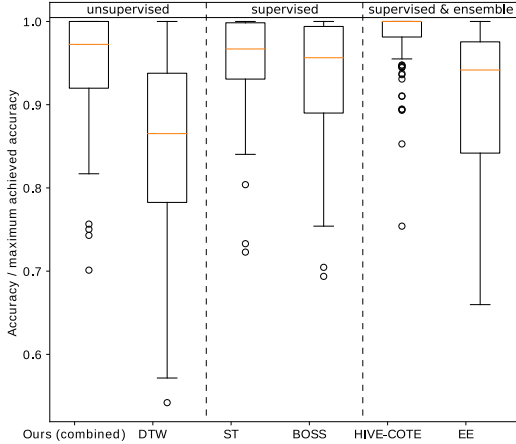 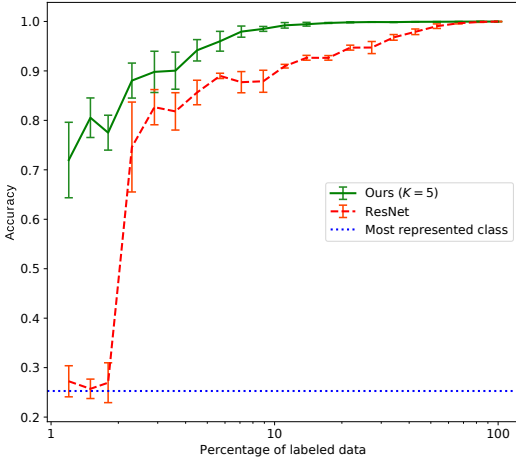

Figure 3: Boxplot of the ratio of the accuracy versus maximum achieved accuracy (higher is better) for compared methods on the first 85 UCR datasets.

Figure 4: Accuracy of ResNet and our method with respect to the ratio of labeled data on TwoPatterns. Error bars correspond to the standard deviation over five runs per point for each method.

training procedure encourages representations of different time series to be separable, observing the classification performance of a simple SVM on these features allows to check their quality (Wu et al., 2018a). Using SVMs also allows, when the encoder is trained, an *efficient* training both in terms of time (training is a matter of minutes in most cases) and space.

As $K$ has a significant impact on the performance, we present a *combined* version of our method, where representations computed by encoders trained with different values of $K$ (see Section S2 for more details) are concatenated. This enables our learned representations with different parameters to complement each other, and to remove some noise in the classification scores.

### 5.1.1 Univariate Time Series

We present accuracy scores for all 128 datasets of the new iteration of the UCR archive (Dau et al., 2018), which is a standard set of varied univariate datasets. We report in Table 1 scores for only some UCR datasets, while scores for all datasets are reported in the supplementary material, Section S3.

We first compare our scores to the two concurrent methods of this work, TimeNet (Malhotra et al., 2017) and RWS (Wu et al., 2018a), which are two unsupervised methods also training a simple classifier on top of the learned representations, and reporting their results on a few UCR datasets. We also compare on the first 85 datasets of the archive[5] to the four best classifiers of the supervised state of the art studied by Bagnall et al. (2017): COTE – replaced by its improved version HIVE-COTE (Lines et al., 2018) –, ST (Bostrom & Bagnall, 2015), BOSS (Schäfer, 2015) and EE (Lines & Bagnall, 2015). HIVE-COTE is a powerful ensemble method using many classifiers in a hierarchical voting structure; EE is a simpler ensemble method; ST is based on shapelets and BOSS is a dictionary-based classifier.[6] We also add DTW (one-nearest-neighbor classifier with DTW as measure) as a baseline. HIVE-COTE includes ST, BOSS, EE and DTW in its ensemble, and is thus expected to outperform them. Additionally, we compare our method to the ResNet method of Wang et al. (2017), which is the best supervised neural network method studied in the review of Ismail Fawaz et al. (2019).

**Performance.** Comparison with the unsupervised state of the art (Section S3, Table S3 of the supplementary material), indicates that our method *consistently matches or outperforms both unsupervised methods* TimeNet and RWS (on 11 out of 12 and 10 out of 11 UCR datasets), showing its

Table 1: Accuracy scores of variants of our method compared with other supervised and unsupervised methods, on some UCR datasets. Results for the whole archive are available in the supplementary material, Section S3, Tables S1, S2 and S4. Bold and underlined scores respectively indicate the best and second-best (when there is no tie for first place) performing methods.

| Dataset | Unsupervised | | | | | Supervised | | | |
| | Ours | | | | DTW | ST | BOSS | Ensemble | |
| | $K = 5$ | $K = 10$ | Combined | FordA | | | | HIVE-COTE | EE |
| --- | --- | --- | --- | --- | --- | --- | --- | --- | --- |
| DiatomSizeReduction | **0.993** | 0.984 | **0.993** | 0.974 | 0.967 | 0.925 | 0.931 | 0.941 | 0.944 |
| ECGFiveDays | 1 | 1 | 1 | 1 | 1 | 0.984 | 1 | 1 | 0.82 |
| FordB | 0.781 | 0.793 | 0.81 | 0.798 | 0.62 | 0.807 | 0.711 | **0.823** | 0.662 |
| Ham | 0.657 | **0.724** | 0.695 | 0.533 | 0.467 | 0.686 | 0.667 | 0.667 | 0.571 |
| Phoneme | 0.249 | 0.276 | 0.289 | 0.196 | 0.228 | 0.321 | 0.265 | **0.382** | 0.305 |
| SwedishLeaf | 0.925 | 0.914 | 0.931 | 0.925 | 0.792 | 0.928 | 0.922 | **0.954** | 0.915 |

performance. Unlike our work, code and full results on the UCR archive are not provided for these methods, hence the incomplete results.

When comparing to the supervised non-neural-network state of the art, we observe (see Figures S2 and S3 in the supplementary material) that our method is globally the second-best one (with average rank 2.92), only beaten by HIVE-COTE (1.71) and equivalent to ST (2.95). Thus, our unsupervised method beats several recognized supervised classifier, and is only preceded by a powerful ensemble method, which was expected since the latter takes advantage of numerous classifiers and data representations. Additionally, Figure 3 shows that our method has the second-best median for the ratio of accuracy over maximum achieved accuracy, behind HIVE-COTE and above ST. Finally, results reported from the study of Ismail Fawaz et al. (2019) for the fully supervised ResNet (Section S3, Table S3 of the supplementary material) show that it expectedly outperforms our method on 63% out of 71 UCR datasets.[7] Overall, our method achieves remarkable performance as it *is close to the best supervised neural network*, *matches the second-best studied non-neural-network supervised method*, and, in particular, is *at the level of the best performing method included in HIVE-COTE*.[8]

**Sparse labeling.** Taking advantage of their unsupervised training, we show that our representations can be successfully used on sparsely labeled datasets compared to supervised methods, since only the SVM is restricted to be learned on the small portion of labeled data. Figure 4 shows that an SVM trained on our representations of a randomly chosen labeled set *consistently outperforms the supervised neural network ResNet* trained on a labeled set of the same size, especially when the percentage of labeled data is small. For example, with only 1.5% of labeled data, we achieve an accuracy of 81%, against only 26% for ResNet, equivalent to a random classifier. Moreover, we exceed 99% of accuracy starting from 11% of labeled data, while ResNet only achieves this level of accuracy with more than 50% of labeled data. This shows the relevance of our method in semi-supervised settings, compared to fully supervised methods.

**Representations metric space.** Besides being suitable for classification purposes, the learned representations may also be used to define a meaningful measure between time series. Indeed, we train, instead of an SVM, a one-nearest-neighbor classifier with respect to the $\ell_2$ distance on the same representations, and compare it to DTW, which uses the same classifier on the raw time series. As shown in S3, this version of our method outperforms DTW on 66% of the UCR datasets, showing the advantage of the learned representations even in a non-parametric classification setting. We also include quantitative experiments to assess the usefulness of comparing time series using the $\ell_2$ distance between their representations with dimensionality reduction (Figure 5) and clustering (Section 5.2 and Figure 6) visualizations.

**Transferability.** We include in the comparisons the classification accuracy for each dataset of an SVM trained on this dataset using the representations computed by an encoder, which was trained *on another dataset* (FordA, with $K = 5$), to test the transferability of our representations. We observe

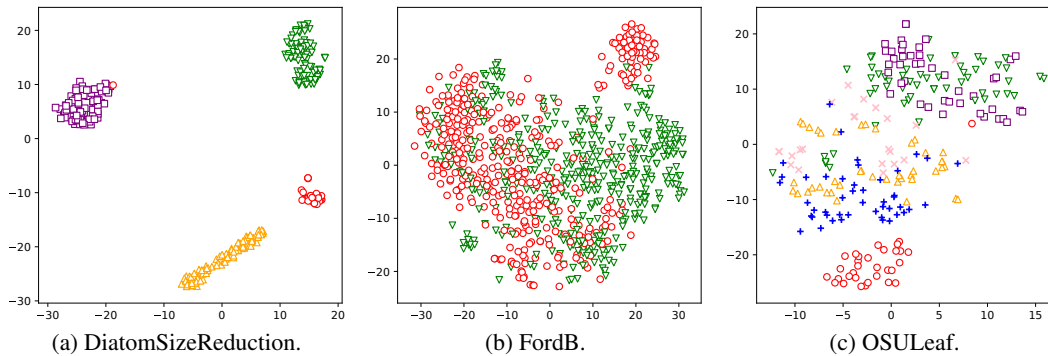

|(a) DiatomSizeReduction.|(b) FordB.|(c) OSULeaf.|

Figure 5: Two-dimensional t-SNE (Maaten & Hinton, 2008) with perplexity 30 of the learned representations of three UCR test sets. Elements classes are distinguishable using their respective marker shapes and colors.

that the scores achieved by this SVM trained on transferred representations are close to the scores reported when the encoder is trained on the same dataset as the SVM, showing the *transferability* of our representations from a dataset to another, and from time series to other time series *with different lengths*. More generally, this transferability and the performance of simple classifiers on the representations we learn indicate that they are *universal* and *easy to make use of*.

### 5.1.2 Multivariate Time Series

To complement our evaluation on the UCR archive which exclusively contains univariate series, we evaluate our method on multivariate time series. This can be done by simply changing the number of input filters of the first convolutional layer of the proposed encoder. We test our method on all 30 datasets of the newly released UEA archive (Bagnall et al., 2018). Full accuracy scores are presented in the supplementary material, Section S4, Table S5.

The UEA archive has been designed as a first attempt to provide a standard archive for multivariate time series classification such as the UCR one for univariate series. As it has only been released recently, we could not compare our method to state-of-the-art classifiers for multivariate time series. However, we provide a comparison with $DTW_D$ as baseline using results provided by Bagnall et al. (2018). $DTW_D$ (dimension-Dependent DTW) is a possible extension of DTW in the multivariate setting, and is the best baseline studied by Bagnall et al. (2018). Overall, our method matches or outperforms $DTW_D$ on 69% of the UEA datasets, which indicates good performance. As this archive is destined to grow and evolve in the future, and without further comparisons, no additional conclusion can be drawn.

### 5.2 Evaluation on Long Time Series

We show the *applicability* and *scalability* of our method on *long* time series without labeling for regression tasks, which could correspond to an industrial application and complements the performed tests on the UCR and UEA archives, whose datasets mostly contain short time series.

The Individual Household Electric Power Consumption (IHEPC) dataset from the UCI Machine Learning Repository (Dheeru & Karra Taniskidou, 2017) is a single time series of length $2\,075\,259$ monitoring the minute-averaged electricity consumption of one French household for four years. We split this time series into train (first $5 \times 10^5$ measurements, approximately a year) and test (remaining measurements). The encoder is trained over the train time series on a single Nvidia Tesla P100 GPU in no more than a few hours, showing that our training procedure is *scalable* to long time series.

We consider the learned encoder on two regression tasks involving two different input scales. We compute, for each time step of the time series, the representations of the last window corresponding to a day ($1\,440$ measurements) and a quarter ($12 \cdot 7 \cdot 1\,440$ measurements) *using the same encoder*. An example of application of the day-long representations is shown in Figure 6. The considered tasks consist in, for each time step, predicting the discrepancy between the mean value of the series

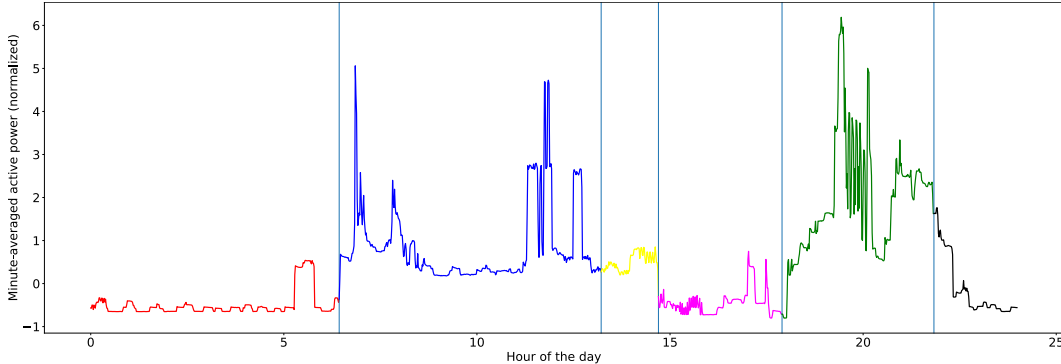

Figure 6: Minute-averaged electricity consumption for a single day, with respect to the hour of the day. Vertical lines and colors divide the day into six clusters, obtained with $k$-means clustering based on representations computed on a day-long sliding window. The clustering divides the day in meaningful portions (night, morning, afternoon, evening).

for the next period (either a day or quarter) and the one for the previous period. We compare linear regressors, trained using gradient descent, to minimize the mean squared error between the prediction and the target, applied either on the raw time series or the previously computed representations.

Results and execution times on an Nvidia Titan Xp GPU are presented in Table 2.[9] On *both scales of inputs*, our representations induce only a slightly degraded performance but provide a *large efficiency improvement*, due to their small size compared to the raw time series. This shows that a single encoder trained to minimize our time-based loss is able to output representations for different scales of input lengths that are also helpful for other tasks than classification, corroborating their *universality*.

Table 2: Results obtained on the IHEPC dataset.

| Task | Metric | Representations | Raw values |
|------|--------|-----------------|------------|
| Day | Test MSE | $\mathbf{8.92 \times 10^{-2}}$ | $\mathbf{8.92 \times 10^{-2}}$ |
|     | Wall time | **12s** | 3min 1s |
| Quarter | Test MSE | $7.26 \times 10^{-2}$ | $\mathbf{6.26 \times 10^{-2}}$ |
|         | Wall time | **9s** | 1h 40min 15s |

## 6 Conclusion

We present an unsupervised representation learning method for time series that is scalable and produces high-quality and easy-to-use embeddings. They are generated by an encoder formed by dilated convolutions that admits variable-length inputs, and trained with an efficient triplet loss using novel time-based negative sampling for time series. Conducted experiments show that these representations are universal and can easily and efficiently be used for diverse tasks such as classification, for which we achieve state-of-the-art performance, and regression.

**Acknowledgements**

We would like to acknowledge Patrick Gallinari, Sylvain Lamprier, Mehdi Lamrayah, Etienne Simon, Valentin Guiguet, Clara Gainon de Forsan de Gabriac, Eloi Zablocki, Antoine Saporta, Edouard Delasalles, Sidak Pal Singh, Andreas Hug, Jean-Baptiste Cordonnier, Andreas Loukas and François Fleuret for helpful comments and discussions. We thank as well our anonymous reviewers for their constructive suggestions, Liljefors et al. (2019) for their extensive and positive reproducibility report on our work, and all contributors to the datasets and archives we used for this project (Dau et al., 2018; Bagnall et al., 2018; Dheeru & Karra Taniskidou, 2017). We acknowledge financial support from the SFA-AM ETH Board initiative, the LOCUST ANR project (ANR-15-CE23-0027) and CLEAR (Center for LEArning and data Retrieval, joint laboratory with Thales[10]).

## Footnotes

[2]I.e., a subsequence of a time series composed by consecutive time steps of this time series.

[3]We used this optimization for multivariate or long (with length higher than 10 000) time series.

[4]https://github.com/White-Link/UnsupervisedScalableRepresentationLearningTimeSeries.

[5]The new UCR archive includes 43 new datasets on which no reproducible results of state-of-the-art methods have been produced yet. Still, we provide complete results for our method on these datasets in the supplementary material, Section S3, Table S4, along with those of DTW, the only other method for which they were available.

[6]While ST and BOSS are also ensembles of classifiers, we chose not to qualify both of them as ensembles since their ensemble only includes variations of the same novel classification method.

[7]Those results are incomplete as Ismail Fawaz et al. (2019) performed their experiments on the old version of the archive, whereas ours are performed on its most recent release where some datasets were changed.

[8]Our method could be included in HIVE-COTE, which could improve its performance, but this is beyond the scope of this work and requires technical work, as HIVE-COTE is implemented in Java and ours in Python.

[9]While acting on representations of the same size, the quarterly linear regressor is slightly faster than the daily one because the number of quarters in the considered time series is smaller than the number of days.

[10]https://www.thalesgroup.com.

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
