[Supplementary Material]

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

# Appendices

In these appendices, we provide our detailed training procedure for classification tasks, choices of hyperparameters, as well as the full experimental results of our method, compared to those of concurrent methods. Section S1 explains and discusses the exact training process for classification tasks. Section S2 details the choices of hyperparameters in all presented experiments. Section S3 reports accuracy scores of all variants of our method on the whole UCR archive (Dau et al., 2018), as well as comparisons with concurrent methods, when available. Section S4 provides accuracy scores for our method on the whole UEA archive (Bagnall et al., 2018). Finally, Section S5 discusses the importance of the choice of encoder by providing results obtained with our training procedure using an LSTM as encoder.

## S1 Training Details

### S1.1 Input Preprocessing

We preprocess datasets of the UCR archive that were not already normalized, as well as the IHEPC dataset, so that the set of time series values for each dataset has zero mean and unit variance. For each UEA dataset, each dimension of the time series was preprocessed independently from the other dimensions by normalizing in the same way its mean and variance.

### S1.2 SVM Training

In order to train an SVM on the computed representations of the elements of the train set, we perform a hyperparameter optimization for the penalty $C$ of the error term of the SVM by cross-validating it over the representations of the train set, thus only using the train labels. Note that if the train set or the number of training samples per class are too small, we choose a penalty $C = \infty$ for the SVM (which corresponds to no regularization).

### S1.3 Behavior of the Learned Representations through Training

**Classification accuracy evolution during training.** As shown in Figure S1, our unsupervised training clearly makes the classification accuracy of the trained SVM increase with the number of optimization steps.

**Numerical stability.** The *Risk* $R$ is defined as the expectation (taken over the random selection of the sequences $(\boldsymbol{x}^{\text{ref}}, \boldsymbol{x}^{\text{pos}}, \boldsymbol{x}^{\text{neg}})$) of the loss defined in Equation (1). This risk may decrease if all the representations $\boldsymbol{f}(\cdot, \boldsymbol{\theta})$ are scaled by a positive large number. For example, if for some $\boldsymbol{\theta}_0$, for (almost surely) any sequences $(\boldsymbol{x}^{\text{ref}}, \boldsymbol{x}^{\text{pos}}, \boldsymbol{x}^{\text{neg}})$, $\boldsymbol{f}(\boldsymbol{x}^{\text{ref}}, \boldsymbol{\theta}_0)^{\top} \boldsymbol{f}(\boldsymbol{x}^{\text{pos}}, \boldsymbol{\theta}_0) \geq 0$ and $\boldsymbol{f}(\boldsymbol{x}^{\text{ref}}, \boldsymbol{\theta}_0)^{\top} \boldsymbol{f}(\boldsymbol{x}^{\text{neg}}, \boldsymbol{\theta}_0) \leq 0$, then

$$R(\lambda, \boldsymbol{\theta}_0) := \mathbb{E}_{\boldsymbol{x}^{\text{ref}}, \boldsymbol{x}^{\text{pos}}, \boldsymbol{x}^{\text{neg}}} \left[ -\log\left(\sigma\left(\lambda^2 \boldsymbol{f}(\boldsymbol{x}^{\text{ref}}, \boldsymbol{\theta}_0)^{\top} \boldsymbol{f}(\boldsymbol{x}^{\text{pos}}, \boldsymbol{\theta}_0)\right)\right) \right.$$
$$\left. -\log\left(\sigma\left(-\lambda^2 \boldsymbol{f}(\boldsymbol{x}^{\text{ref}}, \boldsymbol{\theta}_0)^{\top} \boldsymbol{f}(\boldsymbol{x}^{\text{neg}}, \boldsymbol{\theta}_0)\right)\right) \right] \quad (2)$$

is a decreasing function of $\lambda$, thus $\lambda$ could diverge to infinity in order to minimize the loss. In other words, the parameters in $\boldsymbol{\theta}_0$ corresponding to the last linear layer could be linearly scaled up, and representations would "explode" (their norm would always increase through training). Such a phenomenon is not observed in practice, as the mean representation Euclidean norm lies around 20. There are two possible explanations for that: either the condition above is not satisfied (more generally, the loss is not reduced by increasing the representations) or the use of the sigmoid function, that has vanishing gradients, results in an increase of the representations that is too slow to be observed, or negligible with respect to other weight updates during optimization.

Figure S1: Evolution of the test accuracy during the training of the encoder on the CricketX dataset from the UCR archive (with $K = 10$), with respect to the number of completed epochs. The test labels were only used for monitoring purposes and the test accuracy was computed after each mini-batch optimization. The vertical line marks the epoch at which 2000 optimization steps were performed, at which point training is stopped in our tests. Test accuracy clearly increases during training.

## S2 Hyperparameters

### S2.1 Influence of $K$

As mentioned in Section 5, $K$ can have a significant impact on the performance of the encoder. We notably observed that $K = 1$ leads to statistically significantly lower scores compared to scores obtained when trained with $K > 1$ on the UCR datasets, justifying the use of several negative examples during training. We did not observe any clear statistical difference between other values of $K$ on the whole archive; however, we noticed important differences between different values of $K$ when studying individual datasets. Therefore, we chose to combine several encoders trained with different values of $K$ in order to avoid selecting it as a fixed hyperparameter.

### S2.2 Detailed Choices of Hyperparameters

We train our models with the following parameters for time series classification. Note that *no hyperparameter optimization* was performed on the encoder hyperparameters.

- Optimizer: Adam (Kingma & Ba, 2015) with learning rate $\alpha = 0.001$ and decay rates $\beta = (0.9, 0.999)$.
- SVM: penalty $C \in \left\{ 10^i \mid i \in [\![-4, 4]\!] \right\} \cup \{\infty\}$.
- Encoder training:
  - number of negative samples: $K \in \{1, 2, 5, 10\}$ for univariate time series, $K \in \{5, 10, 20\}$ for multivariate ones;
  - batch size: 10;
  - number of optimizations steps: 2000 for $K \geq 10$ (i.e., 20 epochs for a dataset of size 1000), 1500 otherwise.
- Architecture:
  - number of channels in the intermediary layers of the causal network: 40;
  - number of layers (depth of the causal network): 10;
  - kernel size of all convolutions: 3;
  - negative slope of the leaky ReLU activation: 0.01;
  - number of output channels of the causal network (before max pooling): 320;
  - dimension of the representations: 160.

Figure S2: Critical difference diagram of the average ranks of the compared classifiers for the Nemenyi test, obtained with Orange (Demšar et al., 2013).

Figure S3: Distribution of ranks of compared methods for the first 85 UCR datasets.

For the Individual Household Electric Power Consumption dataset, changes are the following:

- number of negative samples: $K = 10$;
- batch size: $1$;
- number of optimization steps: $400$;
- number of channels in the intermediary layers of the causal network: $30$;
- number of output channels of the causal network (before max pooling): $160$;
- dimension of the representations: $80$.

## S3 Univariate Time Series

Full results corresponding to the first 85 UCR datasets for our method are presented in Table S1, while comparisons with DTW, ST, BOSS, HIVE-COTE and EE are shown in Figures S2 and S3 and Table S2,[S1] and comparisons with ResNet,[S2] TimeNet and RWS are shown in Table S3. Table S4 compiles the results of our method and of DTW[S3] for the newest 43 UCR datasets (except DodgerLoopDay, DodgerLoopGame and DodgerLoopWeekend which contain missing values).

---

[S1]Scores taken from http://www.timeseriesclassification.com/singleTrainTest.csv.

[S2]Scores taken from https://github.com/hfawaz/dl-4-tsc/blob/master/results/results-uea.csv (first iteration).

[S3]Scores taken from https://www.cs.ucr.edu/~eamonn/time_series_data_2018/.

**Standard deviation.** All UCR datasets are provided with a unique train / test split that we used in our experiments. Compared techniques (DTW, ST, BOSS, HIVE-COTE and EE) were also tested on 100 random train / test splits of these datasets by Bagnall et al. (2017) to produce a strong state-of-the-art evaluation, but we did not perform similar resamples as this is beyond the scope of this work and would require much more computations. Note that the scores for these methods used in this article are the ones corresponding to the original train / test split of the datasets.

As our method is based on random sampling, the reported scores may vary depending on the random seed. While we do not report standard deviation, the large number of tested datasets prevents large statistical error in the global evaluation of our method. The order of magnitude of accuracy variation between different runs of the combined version of our method is below 0.01 (for instance, on four different runs, the corresponding standard variations for, respectively, datasets DiatomSizeReduction, CricketX and UWaveGestureLibraryX are 0.0056, 0.0091 and 0.0053).

Table S1: Accuracy scores of variants of our method on the first 85 UCR datasets. "Combined (1-NN)" corresponds to learning a one-nearest-neighbor classifier, instead of an SVM, on the combined representations. Bold scores indicate the best performing method.

| | Unsupervised | | | | | | |
| Dataset | Ours | | | | | | |
| | $K = 1$ | $K = 2$ | $K = 5$ | $K = 10$ | Combined | Combined (1-NN) | FordA |
|---|---|---|---|---|---|---|---|
| Adiac | 0.734 | 0.711 | 0.703 | 0.675 | 0.716 | 0.645 | **0.76** |
| ArrowHead | **0.869** | 0.829 | 0.754 | 0.766 | 0.829 | 0.817 | 0.817 |
| Beef | **0.733** | 0.567 | 0.7 | 0.667 | 0.7 | 0.6 | 0.667 |
| BeetleFly | **0.9** | 0.8 | **0.9** | 0.8 | **0.9** | 0.8 | 0.8 |
| BirdChicken | 0.7 | 0.8 | **0.9** | 0.85 | 0.8 | 0.75 | **0.9** |
| Car | 0.75 | 0.767 | 0.633 | 0.833 | 0.817 | 0.8 | **0.85** |
| CBF | 0.982 | 0.991 | 0.99 | 0.983 | **0.994** | 0.978 | 0.988 |
| ChlorineConcentration | 0.719 | 0.747 | 0.739 | 0.749 | **0.782** | 0.588 | 0.688 |
| CinCECGTorso | 0.702 | **0.747** | 0.682 | 0.713 | 0.74 | 0.693 | 0.638 |
| Coffee | 0.964 | **1** | **1** | **1** | **1** | **1** | **1** |
| Computers | **0.688** | 0.644 | 0.676 | 0.664 | 0.628 | 0.604 | 0.648 |
| CricketX | 0.736 | 0.71 | 0.7 | 0.713 | **0.777** | 0.741 | 0.682 |
| CricketY | 0.682 | 0.664 | 0.695 | 0.728 | **0.767** | 0.664 | 0.667 |
| CricketZ | 0.721 | 0.71 | 0.726 | 0.708 | **0.764** | 0.723 | 0.656 |
| DiatomSizeReduction | 0.99 | 0.987 | **0.993** | 0.984 | **0.993** | 0.967 | 0.974 |
| DistalPhalanxOutlineCorrect | 0.761 | 0.746 | **0.775** | **0.775** | 0.768 | 0.757 | 0.764 |
| DistalPhalanxOutlineAgeGroup | 0.719 | **0.748** | 0.719 | 0.727 | 0.734 | 0.683 | 0.727 |
| DistalPhalanxTW | **0.698** | 0.676 | 0.662 | 0.676 | 0.676 | 0.669 | 0.669 |
| Earthquakes | **0.748** | **0.748** | **0.748** | **0.748** | **0.748** | 0.64 | **0.748** |
| ECG200 | 0.87 | 0.9 | 0.86 | **0.94** | 0.9 | 0.85 | 0.83 |
| ECG5000 | 0.939 | 0.939 | 0.937 | 0.933 | 0.936 | 0.925 | **0.94** |
| ECGFiveDays | **1** | **1** | **1** | **1** | **1** | 0.999 | **1** |
| ElectricDevices | 0.709 | 0.7 | 0.712 | 0.707 | **0.732** | 0.646 | 0.676 |
| FaceAll | 0.764 | **0.81** | 0.733 | 0.786 | 0.802 | 0.75 | 0.734 |
| FaceFour | 0.807 | 0.864 | 0.795 | **0.92** | 0.875 | 0.864 | 0.83 |
| FacesUCR | 0.885 | 0.871 | 0.886 | 0.884 | **0.918** | 0.86 | 0.835 |
| FiftyWords | 0.763 | 0.734 | 0.727 | 0.732 | **0.78** | 0.716 | 0.745 |
| Fish | 0.903 | 0.909 | 0.891 | 0.891 | 0.88 | 0.823 | **0.96** |
| FordA | 0.923 | 0.922 | 0.927 | 0.928 | **0.935** | 0.863 | 0.927 |
| FordB | 0.786 | 0.788 | 0.781 | 0.793 | **0.81** | 0.748 | 0.798 |
| GunPoint | 0.953 | 0.987 | 0.987 | 0.98 | **0.993** | 0.833 | 0.987 |
| Ham | 0.648 | 0.686 | 0.657 | **0.724** | 0.695 | 0.533 | 0.533 |
| HandOutlines | **0.922** | 0.919 | 0.908 | **0.922** | **0.922** | 0.832 | 0.919 |
| Haptics | 0.445 | 0.435 | 0.432 | **0.49** | 0.455 | 0.354 | 0.474 |
| Herring | **0.609** | 0.594 | 0.578 | 0.594 | 0.578 | 0.563 | 0.578 |
| InlineSkate | 0.425 | 0.429 | 0.427 | 0.371 | **0.447** | 0.4 | 0.444 |
| InsectWingbeatSound | 0.61 | 0.592 | 0.617 | 0.597 | **0.623** | 0.506 | 0.599 |
| ItalyPowerDemand | 0.94 | 0.927 | 0.928 | **0.954** | 0.925 | 0.942 | 0.929 |
| LargeKitchenAppliances | 0.797 | 0.827 | 0.843 | 0.789 | **0.848** | 0.757 | 0.765 |
| Lightning2 | 0.869 | 0.836 | 0.852 | 0.869 | **0.918** | 0.885 | 0.787 |
| Lightning7 | 0.795 | **0.822** | **0.822** | 0.795 | 0.795 | 0.795 | 0.74 |
| Mallat | 0.962 | 0.931 | 0.947 | 0.951 | **0.964** | 0.944 | 0.916 |
| Meat | 0.917 | 0.867 | 0.867 | **0.95** | **0.95** | 0.9 | 0.867 |
| MedicalImages | 0.738 | 0.768 | 0.77 | 0.75 | **0.784** | 0.693 | 0.725 |
| MiddlePhalanxOutlineCorrect | 0.749 | 0.818 | 0.777 | **0.825** | 0.814 | 0.722 | 0.787 |
| MiddlePhalanxOutlineAgeGroup | 0.617 | **0.662** | 0.656 | 0.656 | 0.656 | 0.506 | 0.623 |
| MiddlePhalanxTW | 0.604 | **0.61** | **0.61** | 0.591 | **0.61** | 0.513 | 0.584 |
| MoteStrain | **0.875** | 0.854 | 0.867 | 0.851 | 0.871 | 0.853 | 0.823 |
| NonInvasiveFetalECGThorax1 | 0.912 | 0.911 | 0.904 | 0.878 | 0.91 | 0.798 | **0.925** |
| NonInvasiveFetalECGThorax2 | 0.925 | 0.925 | 0.918 | 0.919 | 0.927 | 0.82 | **0.93** |
| OliveOil | 0.867 | 0.833 | 0.867 | 0.867 | **0.9** | 0.833 | **0.9** |
| OSULeaf | 0.719 | 0.694 | 0.793 | 0.76 | **0.831** | 0.636 | 0.736 |
| PhalangesOutlinesCorrect | **0.807** | 0.796 | 0.795 | 0.784 | 0.801 | 0.752 | 0.784 |
| Phoneme | 0.264 | 0.265 | 0.249 | 0.276 | **0.289** | 0.197 | 0.196 |
| Plane | 0.99 | **1** | 0.99 | 0.99 | 0.99 | **1** | 0.981 |
| ProximalPhalanxOutlineCorrect | **0.869** | 0.863 | 0.856 | 0.859 | 0.859 | 0.801 | **0.869** |
| ProximalPhalanxOutlineAgeGroup | 0.849 | **0.859** | 0.844 | 0.844 | 0.854 | 0.805 | 0.839 |
| ProximalPhalanxTW | **0.824** | 0.815 | 0.761 | 0.771 | **0.824** | 0.717 | 0.785 |
| RefrigerationDevices | 0.531 | 0.507 | 0.547 | 0.515 | 0.517 | 0.475 | **0.555** |
| ScreenType | 0.408 | 0.411 | **0.427** | 0.416 | 0.413 | 0.389 | 0.384 |
| ShapeletSim | **0.894** | 0.5 | 0.628 | 0.672 | 0.817 | 0.772 | 0.517 |
| ShapesAll | 0.847 | 0.84 | 0.857 | 0.848 | **0.875** | 0.823 | 0.837 |
| SmallKitchenAppliances | 0.68 | 0.667 | 0.715 | 0.677 | 0.715 | 0.619 | **0.731** |
| SonyAIBORobotSurface1 | **0.93** | 0.89 | 0.85 | 0.902 | 0.897 | 0.825 | 0.84 |
| SonyAIBORobotSurface2 | 0.885 | 0.933 | 0.928 | 0.889 | **0.934** | 0.885 | 0.832 |
| StarLightCurves | 0.96 | 0.966 | 0.958 | 0.964 | 0.965 | 0.893 | **0.968** |
| Strawberry | 0.951 | 0.946 | **0.954** | **0.954** | 0.946 | 0.903 | 0.946 |
| SwedishLeaf | 0.907 | 0.925 | 0.925 | 0.914 | **0.931** | 0.891 | 0.925 |
| Symbols | 0.937 | 0.931 | **0.965** | 0.963 | **0.965** | 0.933 | 0.945 |
| SyntheticControl | 0.98 | 0.983 | **0.987** | **0.987** | 0.983 | 0.977 | 0.977 |

Table S1: Accuracy scores of variants of our method on the first 85 UCR datasets. "Combined (1-NN)" corresponds to learning a one-nearest-neighbor classifier, instead of an SVM, on the combined representations. Bold scores indicate the best performing method.

| | Unsupervised | | | | | | |
|---|---|---|---|---|---|---|---|
| Dataset | Ours | | | | | | |
| | $K=1$ | $K=2$ | $K=5$ | $K=10$ | Combined | Combined (1-NN) | FordA |
| ToeSegmentation1 | 0.868 | **0.961** | 0.93 | 0.939 | 0.952 | 0.851 | 0.899 |
| ToeSegmentation2 | 0.869 | 0.892 | 0.838 | **0.9** | 0.885 | **0.9** | **0.9** |
| Trace | **1** | **1** | **1** | 0.99 | **1** | **1** | **1** |
| TwoLeadECG | 0.996 | 0.991 | 0.996 | **0.999** | 0.997 | 0.988 | 0.993 |
| TwoPatterns | 0.998 | **1** | **1** | 0.999 | **1** | 0.998 | 0.992 |
| UWaveGestureLibraryX | 0.795 | 0.791 | 0.806 | 0.785 | **0.811** | 0.762 | 0.784 |
| UWaveGestureLibraryY | 0.716 | 0.717 | 0.702 | 0.71 | **0.735** | 0.666 | 0.697 |
| UWaveGestureLibraryZ | 0.738 | 0.735 | 0.741 | 0.757 | **0.759** | 0.679 | 0.729 |
| UWaveGestureLibraryAll | 0.893 | 0.887 | 0.903 | 0.896 | **0.941** | 0.838 | 0.865 |
| Wafer | 0.991 | **0.995** | 0.993 | 0.992 | 0.993 | 0.987 | **0.995** |
| Wine | 0.704 | 0.815 | 0.852 | 0.815 | **0.87** | 0.5 | 0.685 |
| WordSynonyms | 0.63 | 0.646 | 0.676 | 0.691 | **0.704** | 0.633 | 0.641 |
| Worms | 0.662 | **0.74** | 0.688 | 0.727 | 0.714 | 0.597 | 0.688 |
| WormsTwoClass | 0.753 | 0.766 | 0.74 | 0.792 | **0.818** | 0.805 | 0.753 |
| Yoga | 0.824 | 0.854 | 0.831 | 0.837 | **0.878** | 0.837 | 0.828 |

Table S2: Accuracy scores of the combined version of our method compared with those of DTW (unsupervised), ST and BOSS (supervised) and HIVE-COTE and EE (supervised ensemble methods), on the first 85 UCR datasets (results on the full archive were not available for comparisons). Bold scores indicate the best performing method.

| | Unsupervised | | Supervised | | | |
| | Ours | DTW | ST | BOSS | Ensemble | |
| Dataset | Combined | | | | HIVE-COTE | EE |
|---|---|---|---|---|---|---|
| Adiac | 0.716 | 0.604 | 0.783 | 0.765 | **0.811** | 0.665 |
| ArrowHead | 0.829 | 0.703 | 0.737 | 0.834 | **0.863** | 0.811 |
| Beef | 0.7 | 0.633 | 0.9 | 0.8 | **0.933** | 0.633 |
| BeetleFly | 0.9 | 0.7 | 0.9 | 0.9 | **0.95** | 0.75 |
| BirdChicken | 0.8 | 0.75 | 0.8 | **0.95** | 0.85 | 0.8 |
| Car | 0.817 | 0.733 | **0.917** | 0.833 | 0.867 | 0.833 |
| CBF | 0.994 | 0.997 | 0.974 | 0.998 | **0.999** | 0.998 |
| ChlorineConcentration | **0.782** | 0.648 | 0.7 | 0.661 | 0.712 | 0.656 |
| CinCECGTorso | 0.74 | 0.651 | 0.954 | 0.887 | **0.996** | 0.942 |
| Coffee | **1** | **1** | 0.964 | **1** | **1** | **1** |
| Computers | 0.628 | 0.7 | 0.736 | 0.756 | **0.76** | 0.708 |
| CricketX | 0.777 | 0.754 | 0.772 | 0.736 | **0.823** | 0.813 |
| CricketY | 0.767 | 0.744 | 0.779 | 0.754 | **0.849** | 0.805 |
| CricketZ | 0.764 | 0.754 | 0.787 | 0.746 | **0.831** | 0.782 |
| DiatomSizeReduction | **0.993** | 0.967 | 0.925 | 0.931 | 0.941 | 0.944 |
| DistalPhalanxOutlineCorrect | 0.768 | 0.717 | **0.775** | 0.728 | 0.772 | 0.728 |
| DistalPhalanxOutlineAgeGroup | 0.734 | **0.77** | **0.77** | 0.748 | 0.763 | 0.691 |
| DistalPhalanxTW | 0.676 | 0.59 | 0.662 | 0.676 | **0.683** | 0.647 |
| Earthquakes | **0.748** | 0.719 | 0.741 | **0.748** | **0.748** | 0.741 |
| ECG200 | **0.9** | 0.77 | 0.83 | 0.87 | 0.85 | 0.88 |
| ECG5000 | 0.936 | 0.924 | 0.944 | 0.941 | **0.946** | 0.939 |
| ECGFiveDays | **1** | 0.768 | 0.984 | **1** | **1** | 0.82 |
| ElectricDevices | 0.732 | 0.602 | 0.747 | **0.799** | 0.77 | 0.663 |
| FaceAll | 0.802 | 0.808 | 0.779 | 0.782 | 0.803 | **0.849** |
| FaceFour | 0.875 | 0.83 | 0.852 | **1** | 0.955 | 0.909 |
| FacesUCR | 0.918 | 0.905 | 0.906 | 0.957 | **0.963** | 0.945 |
| FiftyWords | 0.78 | 0.69 | 0.705 | 0.705 | 0.809 | **0.82** |
| Fish | 0.88 | 0.823 | **0.989** | **0.989** | **0.989** | 0.966 |
| FordA | 0.935 | 0.555 | 0.971 | 0.93 | **0.964** | 0.738 |
| FordB | 0.81 | 0.62 | 0.807 | 0.711 | **0.823** | 0.662 |
| GunPoint | 0.993 | 0.907 | **1** | **1** | **1** | 0.993 |
| Ham | **0.695** | 0.467 | 0.686 | 0.667 | 0.667 | 0.571 |
| HandOutlines | 0.922 | 0.881 | **0.932** | 0.903 | **0.932** | 0.889 |
| Haptics | 0.455 | 0.377 | **0.523** | 0.461 | 0.519 | 0.393 |
| Herring | 0.578 | 0.531 | 0.672 | 0.547 | **0.688** | 0.578 |
| InlineSkate | 0.447 | 0.384 | 0.373 | **0.516** | 0.5 | 0.46 |
| InsectWingbeatSound | 0.623 | 0.355 | 0.627 | 0.523 | **0.655** | 0.595 |
| ItalyPowerDemand | 0.925 | 0.95 | 0.948 | 0.909 | **0.963** | 0.962 |
| LargeKitchenAppliances | 0.848 | 0.795 | 0.859 | 0.765 | **0.864** | 0.811 |
| Lightning2 | **0.918** | 0.869 | 0.738 | 0.836 | 0.82 | 0.885 |
| Lightning7 | **0.795** | 0.726 | 0.726 | 0.685 | 0.74 | 0.767 |
| Mallat | **0.964** | 0.934 | **0.964** | 0.938 | 0.962 | 0.94 |
| Meat | **0.95** | 0.933 | 0.85 | 0.9 | 0.933 | 0.933 |
| MedicalImages | **0.784** | 0.737 | 0.67 | 0.718 | 0.778 | 0.742 |
| MiddlePhalanxOutlineCorrect | 0.814 | 0.698 | 0.794 | 0.78 | **0.832** | 0.784 |
| MiddlePhalanxOutlineAgeGroup | **0.656** | 0.5 | 0.643 | 0.545 | 0.597 | 0.558 |
| MiddlePhalanxTW | **0.61** | 0.506 | 0.519 | 0.545 | 0.571 | 0.513 |
| MoteStrain | 0.871 | 0.835 | 0.897 | 0.879 | **0.933** | 0.883 |
| NonInvasiveFetalECGThorax1 | 0.91 | 0.79 | **0.95** | 0.838 | 0.93 | 0.846 |
| NonInvasiveFetalECGThorax2 | 0.927 | 0.865 | **0.951** | 0.901 | 0.945 | 0.913 |
| OliveOil | **0.9** | 0.833 | **0.9** | 0.867 | **0.9** | 0.867 |
| OSULeaf | 0.831 | 0.591 | 0.967 | 0.955 | **0.979** | 0.806 |
| PhalangesOutlinesCorrect | 0.801 | 0.728 | 0.763 | 0.772 | **0.807** | 0.773 |
| Phoneme | 0.289 | 0.228 | 0.321 | 0.265 | **0.382** | 0.305 |
| Plane | 0.99 | **1** | **1** | **1** | **1** | **1** |
| ProximalPhalanxOutlineCorrect | 0.859 | 0.784 | **0.883** | 0.849 | 0.88 | 0.808 |
| ProximalPhalanxOutlineAgeGroup | 0.854 | 0.805 | 0.844 | 0.834 | **0.859** | 0.805 |
| ProximalPhalanxTW | **0.824** | 0.761 | 0.805 | 0.8 | 0.815 | 0.766 |
| RefrigerationDevices | 0.517 | 0.464 | **0.581** | 0.499 | 0.557 | 0.437 |
| ScreenType | 0.413 | 0.397 | 0.52 | 0.464 | **0.589** | 0.445 |
| ShapeletSim | 0.817 | 0.65 | 0.956 | **1** | **1** | 0.817 |
| ShapesAll | 0.875 | 0.768 | 0.842 | **0.908** | 0.905 | 0.867 |
| SmallKitchenAppliances | 0.715 | 0.643 | 0.792 | 0.725 | **0.853** | 0.696 |
| SonyAIBORobotSurface1 | **0.897** | 0.725 | 0.844 | 0.632 | 0.765 | 0.704 |
| SonyAIBORobotSurface2 | **0.934** | 0.831 | **0.934** | 0.859 | 0.928 | 0.878 |
| StarLightCurves | 0.965 | 0.907 | 0.979 | 0.978 | **0.982** | 0.926 |
| Strawberry | 0.946 | 0.941 | 0.962 | **0.976** | 0.97 | 0.946 |
| SwedishLeaf | 0.931 | 0.792 | 0.928 | 0.922 | **0.954** | 0.915 |

Table S2: Accuracy scores of the combined version of our method compared with those of DTW (unsupervised), ST and BOSS (supervised) and HIVE-COTE and EE (supervised ensemble methods), on the first 85 UCR datasets (results on the full archive were not available for comparisons). Bold scores indicate the best performing method.

| Dataset | Unsupervised | | Supervised | | | |
| | Ours | DTW | ST | BOSS | Ensemble | |
| | Combined | | | | HIVE-COTE | EE |
| --- | --- | --- | --- | --- | --- | --- |
| Symbols | 0.965 | 0.95 | 0.882 | 0.967 | **0.974** | 0.96 |
| SyntheticControl | 0.983 | 0.993 | 0.983 | 0.967 | **0.997** | 0.99 |
| ToeSegmentation1 | 0.952 | 0.772 | 0.965 | 0.939 | **0.982** | 0.829 |
| ToeSegmentation2 | 0.885 | 0.838 | 0.908 | **0.962** | 0.954 | 0.892 |
| Trace | **1** | **1** | **1** | **1** | **1** | 0.99 |
| TwoLeadECG | **0.997** | 0.905 | **0.997** | 0.981 | 0.996 | 0.971 |
| TwoPatterns | **1** | **1** | 0.955 | 0.993 | **1** | **1** |
| UWaveGestureLibraryX | 0.811 | 0.728 | 0.803 | 0.762 | **0.84** | 0.805 |
| UWaveGestureLibraryY | 0.735 | 0.634 | 0.73 | 0.685 | **0.765** | 0.726 |
| UWaveGestureLibraryZ | 0.759 | 0.658 | 0.748 | 0.695 | **0.783** | 0.724 |
| UWaveGestureLibraryAll | 0.941 | 0.892 | 0.942 | 0.939 | **0.968** | **0.968** |
| Wafer | 0.993 | 0.98 | **1** | 0.995 | 0.999 | 0.997 |
| Wine | **0.87** | 0.574 | 0.796 | 0.741 | 0.778 | 0.574 |
| WordSynonyms | 0.704 | 0.649 | 0.571 | 0.638 | 0.738 | **0.779** |
| Worms | 0.714 | 0.584 | **0.74** | 0.558 | 0.558 | 0.662 |
| WormsTwoClass | 0.818 | 0.623 | **0.831** | **0.831** | 0.779 | 0.688 |
| Yoga | 0.878 | 0.837 | 0.818 | **0.918** | **0.918** | 0.879 |

Table S3: Accuracy scores of the combined version of our method compared with those of ResNet (supervised), TimeNet and RWS (unsupervised), when available. Bold scores indicate the best performing method. "X"'s indicate that a score was reported in the original paper, but was either obtained using transferability or on a reversed train / test split of the dataset, thus not comparable to other results for this dataset.

| Dataset | Unsupervised Ours Combined | Supervised ResNet | Unsupervised TimeNet | Unsupervised RWS |
|---|---|---|---|---|
| Adiac | 0.716 | **0.831** | 0.565 | - |
| ArrowHead | 0.829 | **0.84** | - | - |
| Beef | 0.7 | **0.767** | - | 0.733 |
| BeetleFly | **0.9** | 0.85 | - | - |
| BirdChicken | 0.8 | **0.95** | - | - |
| Car | 0.817 | **0.917** | - | - |
| CBF | **0.994** | 0.989 | - | - |
| ChlorineConcentration | 0.782 | **0.835** | 0.723 | 0.572 |
| CinCECGTorso | 0.74 | **0.838** | - | - |
| Coffee | **1** | **1** | - | - |
| Computers | 0.628 | **0.816** | - | - |
| CricketX | 0.777 | **0.79** | 0.659 | - |
| CricketY | 0.767 | **0.805** | X | - |
| CricketZ | 0.764 | **0.831** | X | - |
| DiatomSizeReduction | **0.993** | 0.301 | - | - |
| DistalPhalanxOutlineCorrect | **0.768** | X | X | - |
| DistalPhalanxOutlineAgeGroup | **0.734** | X | X | - |
| DistalPhalanxTW | **0.676** | X | X | X |
| Earthquakes | **0.748** | X | - | - |
| ECG200 | **0.9** | 0.87 | - | - |
| ECG5000 | **0.936** | 0.935 | 0.934 | 0.933 |
| ECGFiveDays | **1** | 0.99 | X | - |
| ElectricDevices | 0.732 | **0.735** | 0.665 | - |
| FaceAll | 0.802 | **0.855** | - | - |
| FaceFour | 0.875 | **0.955** | - | - |
| FacesUCR | 0.918 | **0.955** | - | - |
| FiftyWords | **0.78** | 0.732 | - | - |
| Fish | 0.88 | **0.977** | - | - |
| FordA | **0.935** | X | X | - |
| FordB | **0.81** | X | X | X |
| GunPoint | **0.993** | **0.993** | - | - |
| Ham | 0.695 | **0.8** | - | - |
| HandOutlines | **0.922** | X | - | X |
| Haptics | 0.455 | **0.516** | - | - |
| Herring | 0.578 | **0.641** | - | - |
| InlineSkate | **0.447** | 0.378 | - | - |
| InsectWingbeatSound | **0.623** | 0.506 | - | 0.619 |
| ItalyPowerDemand | 0.925 | 0.959 | - | **0.969** |
| LargeKitchenAppliances | 0.848 | **0.904** | - | 0.792 |
| Lightning2 | **0.918** | 0.77 | - | - |
| Lightning7 | 0.795 | **0.863** | - | - |
| Mallat | 0.964 | **0.966** | - | 0.937 |
| Meat | 0.95 | **0.983** | - | - |
| MedicalImages | **0.784** | 0.762 | 0.753 | - |
| MiddlePhalanxOutlineCorrect | **0.814** | X | X | X |
| MiddlePhalanxOutlineAgeGroup | **0.656** | X | X | - |
| MiddlePhalanxTW | **0.61** | X | X | - |
| MoteStrain | 0.871 | **0.924** | - | - |
| NonInvasiveFetalECGThorax1 | 0.91 | **0.946** | - | 0.907 |
| NonInvasiveFetalECGThorax2 | 0.927 | **0.944** | - | - |
| OliveOil | **0.9** | 0.867 | - | - |
| OSULeaf | 0.831 | **0.979** | - | - |
| PhalangesOutlinesCorrect | 0.801 | **0.857** | 0.772 | - |
| Phoneme | 0.289 | **0.333** | - | - |
| Plane | 0.99 | **1** | - | - |
| ProximalPhalanxOutlineCorrect | 0.859 | **0.914** | X | 0.711 |
| ProximalPhalanxOutlineAgeGroup | **0.854** | 0.839 | X | X |
| ProximalPhalanxTW | **0.824** | X | X | - |
| RefrigerationDevices | **0.517** | **0.517** | - | - |
| ScreenType | 0.413 | **0.632** | - | - |
| ShapeletSim | 0.817 | **1** | - | - |
| ShapesAll | 0.875 | **0.917** | - | - |
| SmallKitchenAppliances | 0.715 | **0.789** | - | - |
| SonyAIBORobotSurface1 | 0.897 | **0.968** | - | - |
| SonyAIBORobotSurface2 | 0.934 | **0.986** | - | - |
| StarLightCurves | 0.965 | **0.972** | - | - |
| Strawberry | **0.946** | X | X | - |

Table S3: Accuracy scores of the combined version of our method compared with those of ResNet (supervised), TimeNet and RWS (unsupervised), when available. Bold scores indicate the best performing method. "X"'s indicate that a score was reported in the original paper, but was either obtained using transferability or on a reversed train / test split of the dataset, thus not comparable to other results for this dataset.

| Dataset | Unsupervised Ours | Supervised | Unsupervised | |
|---|---|---|---|---|
| | Combined | ResNet | TimeNet | RWS |
| SwedishLeaf | 0.931 | **0.955** | 0.901 | - |
| Symbols | **0.965** | 0.927 | - | - |
| SyntheticControl | 0.983 | **1** | 0.983 | - |
| ToeSegmentation1 | 0.952 | **0.969** | - | - |
| ToeSegmentation2 | 0.885 | **0.915** | - | - |
| Trace | **1** | **1** | - | - |
| TwoLeadECG | 0.997 | **1** | - | - |
| TwoPatterns | **1** | **1** | 0.999 | 0.999 |
| UWaveGestureLibraryX | **0.811** | 0.78 | - | - |
| UWaveGestureLibraryY | **0.735** | 0.675 | - | - |
| UWaveGestureLibraryZ | **0.759** | 0.75 | - | - |
| UWaveGestureLibraryAll | **0.941** | 0.862 | - | - |
| Wafer | 0.993 | **0.998** | 0.994 | 0.993 |
| Wine | **0.87** | 0.611 | - | - |
| WordSynonyms | **0.704** | 0.625 | - | - |
| Worms | **0.714** | X | - | - |
| WormsTwoClass | **0.818** | X | - | - |
| Yoga | **0.878** | 0.857 | 0.866 | - |

Table S4: Accuracy scores of variants of our method and of DTW on the remaining 43 UCR datasets, except DodgerLoopDay, DodgerLoopGame and DodgerLoopWeekend which contain missing values. Bold scores indicate the best performing method.

| | Unsupervised | | | | | | | DTW |
|---|---|---|---|---|---|---|---|---|
| Dataset | Ours | | | | | | | |
| | $K = 1$ | $K = 2$ | $K = 5$ | $K = 10$ | Combined | Combined (1-NN) | FordA | |
| ACSF1 | **0.91** | 0.87 | 0.86 | 0.9 | 0.81 | 0.85 | 0.73 | 0.64 |
| AllGestureWiimoteX | 0.721 | 0.746 | 0.747 | 0.763 | **0.779** | 0.736 | 0.693 | 0.716 |
| AllGestureWiimoteY | 0.741 | 0.744 | 0.759 | 0.726 | **0.793** | 0.756 | 0.713 | 0.729 |
| AllGestureWiimoteZ | 0.687 | 0.697 | 0.691 | 0.723 | **0.763** | 0.716 | 0.71 | 0.643 |
| BME | **0.993** | **0.993** | 0.987 | **0.993** | **0.993** | 0.947 | 0.96 | 0.9 |
| Chinatown | 0.951 | 0.951 | 0.942 | 0.951 | **0.962** | 0.936 | **0.962** | 0.957 |
| Crop | 0.728 | 0.726 | 0.728 | 0.722 | **0.746** | 0.695 | 0.727 | 0.665 |
| EOGHorizontalSignal | 0.552 | 0.566 | 0.536 | **0.605** | 0.588 | 0.522 | 0.47 | 0.503 |
| EOGVerticalSignal | 0.398 | 0.414 | 0.431 | 0.434 | **0.489** | 0.431 | 0.439 | 0.448 |
| EthanolLevel | 0.418 | 0.34 | 0.316 | 0.382 | 0.392 | 0.274 | **0.558** | 0.276 |
| FreezerRegularTrain | 0.986 | 0.988 | 0.979 | 0.956 | 0.955 | 0.963 | **0.992** | 0.899 |
| FreezerSmallTrain | 0.967 | 0.956 | 0.906 | **0.933** | 0.928 | 0.872 | 0.862 | 0.753 |
| Fungi | **1** | **1** | **1** | **1** | **1** | **1** | 0.925 | 0.839 |
| GestureMidAirD1 | **0.638** | 0.577 | 0.592 | 0.608 | 0.615 | 0.546 | 0.608 | 0.569 |
| GestureMidAirD2 | 0.508 | 0.515 | 0.523 | **0.546** | 0.508 | 0.415 | 0.538 | 0.608 |
| GestureMidAirD3 | 0.269 | **0.331** | 0.308 | 0.285 | **0.331** | 0.246 | 0.292 | 0.323 |
| GesturePebbleZ1 | 0.826 | 0.843 | 0.913 | 0.919 | **0.936** | 0.814 | 0.547 | 0.791 |
| GesturePebbleZ2 | 0.861 | 0.873 | 0.88 | **0.899** | 0.88 | 0.791 | 0.538 | 0.671 |
| GunPointAgeSpan | 0.984 | 0.984 | **0.994** | **0.994** | 0.987 | 0.991 | 0.987 | 0.918 |
| GunPointMaleVersusFemale | **1** | **1** | **1** | 0.997 | **1** | 0.994 | **1** | 0.997 |
| GunPointOldVersusYoung | **1** | **1** | **1** | **1** | **1** | **1** | **1** | 0.838 |
| HouseTwenty | **0.95** | 0.933 | 0.916 | 0.933 | **0.95** | 0.924 | 0.882 | 0.924 |
| InsectEPGRegularTrain | **1** | **1** | **1** | **1** | **1** | **1** | **1** | 0.872 |
| InsectEPGSmallTrain | **1** | **1** | **1** | **1** | **1** | **1** | **1** | 0.735 |
| MelbournePedestrian | 0.949 | 0.946 | 0.943 | 0.944 | **0.951** | 0.914 | 0.947 | 0.791 |
| MixedShapesRegularTrain | 0.916 | 0.906 | 0.904 | 0.905 | **0.927** | 0.898 | 0.898 | 0.842 |
| MixedShapesSmallTrain | 0.864 | 0.857 | 0.871 | 0.86 | **0.877** | 0.829 | 0.861 | 0.78 |
| PickupGestureWiimoteZ | 0.72 | **0.8** | 0.78 | 0.74 | 0.78 | 0.72 | 0.74 | 0.66 |
| PigAirwayPressure | 0.385 | 0.452 | **0.51** | **0.51** | 0.486 | 0.332 | 0.317 | 0.106 |
| PigArtPressure | 0.88 | 0.933 | **0.942** | 0.928 | 0.933 | 0.861 | 0.591 | 0.245 |
| PigCVP | 0.404 | 0.548 | 0.62 | **0.788** | 0.712 | 0.385 | 0.534 | 0.154 |
| PLAID | 0.533 | 0.549 | 0.574 | 0.555 | 0.559 | 0.696 | 0.493 | **0.84** |
| PowerCons | **0.961** | 0.939 | 0.9 | 0.9 | 0.928 | 0.894 | 0.933 | 0.878 |
| Rock | 0.62 | 0.62 | 0.58 | 0.58 | **0.68** | 0.5 | 0.54 | 0.6 |
| SemgHandGenderCh2 | 0.845 | 0.852 | 0.873 | 0.89 | **0.902** | 0.863 | 0.84 | 0.802 |
| SemgHandMovementCh2 | 0.711 | 0.649 | 0.7 | **0.789** | 0.784 | 0.709 | 0.516 | 0.584 |
| SemgHandSubjectCh2 | 0.767 | 0.816 | 0.851 | 0.853 | **0.876** | 0.72 | 0.591 | 0.727 |
| ShakeGestureWiimoteZ | 0.92 | 0.92 | **0.94** | 0.92 | **0.94** | 0.86 | 0.9 | 0.86 |
| SmoothSubspace | 0.933 | **0.96** | 0.94 | **0.96** | 0.953 | 0.833 | 0.94 | 0.827 |
| UMD | 0.979 | 0.986 | **0.993** | **0.993** | **0.993** | 0.958 | 0.986 | **0.993** |

## S4    Multivariate Time Series

Full results corresponding to the UEA archive datasets for our method as well as the ones of $DTW_D$ as reported by Bagnall et al. (2018) are presented in Table S5, for the unique train / test split provided in the archive.

Table S5: Accuracy scores of variants of our method on all UEA datasets, compared to $DTW_D$. Bold scores indicate the best performing method.

|  | Unsupervised | | | | |
|---|---|---|---|---|---|
|  | Ours | | | | $DTW_D$ |
| Dataset | $K = 5$ | $K = 10$ | $K = 20$ | Combined | |
| ArticularyWordRecognition | 0.967 | 0.973 | 0.943 | **0.987** | **0.987** |
| AtrialFibrillation | **0.2** | 0.067 | 0.133 | 0.133 | **0.2** |
| BasicMotions | **1** | **1** | **1** | **1** | 0.975 |
| CharacterTrajectories | 0.986 | 0.99 | 0.993 | **0.994** | 0.989 |
| Cricket | 0.958 | 0.972 | 0.972 | 0.986 | **1** |
| DuckDuckGeese | 0.6 | **0.675** | 0.65 | **0.675** | 0.6 |
| EigenWorms | 0.87 | 0.802 | 0.84 | **0.878** | 0.618 |
| Epilepsy | **0.971** | **0.971** | **0.971** | 0.957 | 0.964 |
| Ering | **0.133** | **0.133** | **0.133** | **0.133** | **0.133** |
| EthanolConcentration | 0.289 | 0.251 | 0.205 | 0.236 | **0.323** |
| FaceDetection | 0.522 | 0.525 | 0.513 | 0.528 | **0.529** |
| FingerMovements | 0.55 | 0.49 | **0.58** | 0.54 | 0.53 |
| HandMovementDirection | 0.311 | 0.297 | **0.351** | 0.27 | 0.231 |
| Handwriting | 0.447 | 0.464 | 0.451 | **0.533** | 0.286 |
| Heartbeat | **0.756** | 0.732 | 0.741 | 0.737 | 0.717 |
| InsectWingbeat | 0.159 | 0.158 | 0.156 | **0.16** | - |
| JapaneseVowels | 0.984 | 0.986 | **0.989** | **0.989** | 0.949 |
| Libras | 0.878 | **0.883** | **0.883** | 0.867 | 0.87 |
| LSST | 0.535 | 0.552 | 0.509 | **0.558** | 0.551 |
| MotorImagery | 0.53 | 0.54 | **0.58** | 0.54 | 0.5 |
| NATOPS | 0.933 | 0.917 | 0.917 | **0.944** | 0.883 |
| PEMS-SF | 0.636 | 0.671 | 0.676 | **0.688** | 0.711 |
| PenDigits | **0.985** | 0.979 | 0.981 | 0.983 | 0.977 |
| Phoneme | 0.216 | 0.214 | 0.222 | **0.246** | 0.151 |
| RacketSports | 0.776 | 0.836 | 0.855 | **0.862** | 0.803 |
| SelfRegulationSCP1 | 0.795 | 0.826 | 0.843 | **0.846** | 0.775 |
| SelfRegulationSCP2 | 0.55 | 0.539 | 0.539 | **0.556** | 0.539 |
| SpokenArabicDigits | 0.908 | 0.894 | 0.905 | 0.956 | **0.963** |
| StandWalkJump | 0.333 | **0.4** | 0.333 | **0.4** | 0.2 |
| UWaveGestureLibrary | 0.884 | 0.869 | 0.875 | 0.884 | **0.903** |

## S5 Discussion of the Choice of Encoder

One of the aims of this work is to propose a representation learning method for time series that is scalable. For this reason, and as explained in Section 4, we did not consider using an LSTM as encoder $f$. Nonetheless, we experimented with such an encoder on a small set of UCR datasets in order to get an indication of its performance versus the proposed encoder in this paper. We use the same optimization hyperparameters as those used to train the causal CNN encoder and choose a two-layer LSTM with hidden size 256 in order to compare both networks with similar computational time and memory usage. Corresponding results are compiled in Table S6.

We observe on this restricted set of experiments that not only the proposed encoder outperforms the LSTM encoder, but it does so by a large margin. This indicates that the proposed causal CNN encoder is more adapted to the considered task and training method.

Table S6: Results of our training method combined with the proposed causal CNN encoder on one hand, and with an LSTM encoder on the other hand ($K = 5$). Bold scores indicate the best performing method.

| Dataset | Causal CNN | LSTM |
| --- | --- | --- |
| Adiac | **0.703** | 0.269 |
| Computers | **0.676** | 0.492 |
| CricketX | **0.7** | 0.136 |
| DistalPhalanxTW | **0.662** | **0.662** |
| Earthquakes | **0.748** | **0.748** |
| HandOutlines | **0.908** | 0.646 |
| NonInvasiveFetalECGThorax1 | **0.904** | 0.169 |
| PhalangesOutlinesCorrect | **0.795** | 0.613 |
| RefrigerationDevices | **0.547** | 0.411 |
| UWaveGestureLibraryX | **0.806** | 0.357 |
| Wafer | **0.993** | 0.896 |