[Reviews · NeurIPS 2019]

Reviewer 1



Originality: The methods are not necessarily original - the approach is a pretty straightforward application of triplet embeddings from NLP to time series context. Of course, they did need to map context and positive and negative examples into the time series setting, which they have done. I am curious, though, how the method would perform if the "context" and "positive" examples did not explicitly overlap. For example, if they were simply close to each other in time, either slightly overlapping or adjacent, how would this change the performance. Quality: The paper quality is mediocre overall, though I do like the idea and do want to see it published. Clarity: Can be improved, as the writing is poor at times. Significance: Embeddings for time series is an important problem. This paper does apply a useful technique to embed time series, which to the best of my knowledge has not been done. In that regard, this paper is significant, and the community does need to see these results. That said, there is only one comparison to state of the art existing unsupervised methods in the main paper, DTW, so it is difficult to know how this performs in comparison to other embedding methods like seq2seq. Again, I like the ideas in the paper, I think they could be very useful in many applied areas, but I'm not sure the current version of the paper is NeurIPs quality.

Reviewer 2



Originality: The paper builds on several known ideas (dilated causal convolutions for the encoder, and similar triplet loss ideas from other domains), but the application to learning time series embeddings appears novel (and, as the empirical evaluation seems to demonstrate, effective). Quality: The proposed approach appears technically sounds. The empirical evaluation is extensive and demonstrates several desirable attributes of the proposed embeddings. Clarity: The paper is mostly clear and easy to follow. Some of the details, in particular about the exact model architecture used for the experiments, are relegated to the supplementary material. I was surprised not to see a weighting factor (also depending on K) for the different terms in eq. (1), but looking at the code it seems one was actually used -- this should be described in the paper. It's also not quite clear how the hyperparameters for the experiments where chosen. Significance: The proposed technique, though fairly straightforward and making use of established techniques, appears novel and, especially due to its encouraging results, could lead to further fruitful work in this direction.

Reviewer 3



This paper proposes a model for unsupervised time series modeling. The model consists of an encoder (for subsequences of varying length), a sampling strategy of triplet subsequences, and a loss function called the triplet-loss. The three samples are a reference sequence, a subsequence of the reference called the positive sequence and a negative sequence which is in no relation to the reference or positive sequences. The encoder maps each of the subsequences to their embedding and the triplet loss ensures that the embeddings of the positive sample and the reference are 'similar' to each other, while the positive and negative sample are 'dissimilar' to each other. The claim is that the learned representations capture meaningful features of the time series. Representation learning is an active area of research but somehow embeddings for time series data have not yet enjoyed as much attention in the research community. Time series embeddings are useful because they can map time series data to fixed-length representations which can be used as input features for downstream tasks or for qualitative exploration of the data. In a large-scale empirical study, the authors evaluate the time series embeddings on different tasks; classification, classification with sparse labels, and 'forecasting'. Evaluating time series embeddings is notoriously hard, especially since to quantify their quality a task needs to be defined. But for a given task (e.g. classification) the best model is simply a supervised model. It is intuitively clear that good embeddings are also useful for other tasks (e.g. clustering of the time series) but again it is difficult to quantitatively 'prove' that a specific embedding method would be preferable. Large benchmark time series tasks are only available for classification and not for other tasks. The empirical study of this paper is quite extensive. I would have been curious to see performance on other tasks (though of course I understand there might be no benchmarks). I also would have been curious to see experiments that compare the embeddings of different encoders trained with the triplet loss. This might help understand how much of the performance is due to the encoder architecture and how much is due to the triplet loss being a good objective. Similarly, it would be interesting to see the performance of the proposed encoder trained on a different objective, e.g. autoencoding or the 'forecasting task' from Section 5.3. In Bagnall et al. many of the competitive methods are based on KNN (K-nearest neighbours). Have the authors tried (instead of using an SVM) doing the classification based on KNN? Are the distances in the embedding space more useful than DTW? An advantage would be that one wouldn't need any class-labels during training. I would rate the originality of the work as medium (the encoder architecture and triplet loss come from other work, but how to best combine them requires some thoughts), with significance medium to high, as time series embeddings is an important topic. The work is of good quality (especially the extensive Experiments in a domain that is hard to evaluate) and clearly presented.

[Author Response · NeurIPS 2019]

We would like to thank the reviewers for their comments and suggestions. Below, we address the points raised by the reviewers; we will make sure to update our manuscript accordingly.

## 1 Comparison with Other Unsupervised Methods (R1)

The main paper **does already include** (lines 218-222) a comparison of our method with the two most recent and significant unsupervised methods in the literature, TimeNet [1] and RWS [2], which we consistently outperform.

In particular, TimeNet is a seq2seq method relying on an antoencoding loss and using LSTMs as encoder and decoder. While classically used in NLP, such methods did not receive much attention in the time series community apart from TimeNet, and notably do not scale to long time series (as explained on lines 144-157), unlike ours.

## 2 Additional Experiments (R1, R2, R3)

**Comparison with different losses (R1, R2, R3)** As we focus on scalability, we did not train our encoder with an autoencoding loss. However, we did perform experiments on some datasets with different loss variants. Replacing the word2vec loss with a triplet margin loss and choosing positive samples close to the anchor instead of being included in the anchor segment performs similarly to our triplet loss, but would require more hyperparameter tuning. In the end, we chose to present the loss which is simplest to tune, which is desirable in the context of unsupervised learning.

**Comparison with different encoders (R3)** The encoder architecture can indeed play an important role in the performance of the unsupervised training. As we aimed at achieving scalability of the encoder, we did not provide a comparison between different types of encoders. However, preliminary experiments indicate that using an LSTM to replace the causal CNN resulted in significantly worse results, besides increasing the computational cost of the method. Despite the poor computational scalability of LSTMs to long sequences, we will include additional comparisons between LSTMs and CNNs to shed more light on the importance of the encoder architecture.

**SVMs vs kNNs (R3)** Preliminary tests done by training kNNs on the learned embeddings suggest that they can achieve performance close to the one of SVMs, outperforming DTW. Thus, distances between our learnt embeddings are indeed meaningful. We will add insights on this matter to the paper.

**Clustering (R1)** We will include 2D visualizations of the computed embeddings for some datasets obtained using a dimensionality-reduction technique such as t-SNE.

## 3 Multivariate Time Series (R1)

We **do test** our method on multivariate time series (lines 253-265) using the recently released UEA archive, that is aimed at being used as a reference multivariate archive similarly to the UCR one. To this end, the encoder network is only modified by setting the number of input channels of its first convolution to the number of dimensions of the time series in the dataset. We will clarify this point.

## 4 Hyperparameters (R2)

We provide all hyperparameters values in the supplementary material, and will release all our code. The scaling factor in the code has always been set to one and can thus be discarded in the description of the paper.

Choosing a good set of hyperparameters for an unsupervised method is often challenging since the plurality of downstream tasks are usually supervised. Thus, similarly to what was done in other unsupervised work [2], we chose to keep all hyperparameters constant for each entire dataset archive (i.e., a single set of hyperparameter values for all UCR datasets, and another set for all UEA datasets), without any specific tuning for any particular downstream task. We will clarify these points.

## 5 Writing (R1)

We will proofread the manuscript for language clarity.

## References

[1] P. Malhotra et al. TimeNet: Pre-trained deep recurrent neural network for time series classification. *arXiv:1706.08838*, 2017.

[2] L. Wu et al. Random Warping Series: A random features method for time-series embedding. In *AISTATS*, 2018.


[Meta-Review · NeurIPS 2019]

This paper presents a new technique for time series embedding for multivariate time series of differing lengths. The method encodes the time series with a stack of dilated causal convolutions and uses "triplet-loss" function that has been adapted to the time series domain. Overall, the reviewers found the combination of these pieces novel for the problem the authors were trying to solve. Additionally, the experiments were quite extensive and demonstrated as well as could be asked for that the method performs well and is useful. There were a few criticisms of the evaluation such as comparing to other tasks and using KNN instead of SVMs for classification. The authors seem to have addressed these issues to the reviewers' satisfaction. The authors should incorporate the useful feedback that the reviewers provided in the camera-read version.